# A Web of Challenges: The Therapeutic Struggle to Target NETs in Disease

**DOI:** 10.3390/ijms26104773

**Published:** 2025-05-16

**Authors:** Andre Espiritu, Kim Maree O’Sullivan

**Affiliations:** Department of Medicine, Centre for Inflammatory Diseases, Monash University, Clayton, VIC 3168, Australia; andre.espiritu@monash.edu

**Keywords:** neutrophil extracellular traps, NET-targeted therapies, DNase I, Ninjurin-1, CIT-013

## Abstract

Neutrophil extracellular traps (NETs) play a crucial role in the pathophysiology of many debilitating conditions, including autoimmune diseases, inflammatory diseases, and cancer. As a result, NET-targeted therapies have been investigated in search of effective treatment strategies. Despite promising preclinical findings, clinical translation of NET inhibitors has had limited success. These preclinical studies have faced limitations such as mouse models that inaccurately reflect human disease dynamics, as well as by the complexity of NETs—including their diverse morphology and convoluted pathways to formation relative to pathology. The NET inhibitors themselves have several limitations, including off-target effects and bioavailability issues. The challenges facing NET-targeted therapies reported here may explain what is required to go from bench to bedside successfully.

## 1. Introduction

Innate immunity is a highly conserved biological system that is typically described as the first line of defence against infections and injury. One of the major components of the innate immune system is neutrophils: short-lived granulocytes that are the most abundant type of white blood cells. In humans, these bone marrow-produced leukocytes have a circulating lifespan of 12–18 h, with activation extending this to as long as 48 h [1,2,3,4]. Since their discovery by Max Schultze in 1865, neutrophils were thought to have only two primary functions: phagocytosis and degranulation [5]. Phagocytosis refers to the ingestion of harmful particles or host cellular debris, thereby regulating homeostasis and immunity, and degranulation involves the release of biologically active intracellular proteins with potent antimicrobial and inflammatory effects. In the late 20th century, over a hundred years since the discovery of neutrophils, researchers hinted at a third primary function: the formation of neutrophil extracellular traps (NETs), a unique form of cell death believed to increase neutrophil bactericidal capacity.

In 1993, Takeda et al. reported that tumour necrosis factor (TNF), a potent activator of neutrophils, induced rapid death of these cells [6]. This was in direct contrast with other known activators of neutrophils at the time, such as granulocyte-macrophage colony-stimulating factor (GM-CSF) and interleukin-1 (IL-1), which generally prolonged the neutrophil lifespan by preventing pro-apoptotic signalling pathways. A follow-up study saw a similar acceleration of cell death upon stimulating with the neutrophil activator phorbol 12-myristate 13-acetate (PMA) [7]. However, it was noted that this type of cell death was distinct from the prototypical apoptosis and necrosis. Using electron microscopy, the authors were able to determine that this PMA-induced cell death involved fusion of the nuclear lobules, followed by nuclear swelling and a reduction of chromatin compactness. This nuclear transformation did not coincide with swelling of the entire cell, as is the key differentiator from necrosis. Breaking of the nuclear envelope allowed mixing of the nucleoplasm with intact cytoplasmic contents such as mitochondria, granules, and phagosomes. Finally, the plasma membrane ruptured, and the amalgamated contents were released extracellularly. This process would later be coined NETs or NETosis in 2004 by Brinkmann et al., who are credited with its discovery (Figure 1) [8].

Although they were not technically the first researchers to publish findings about NETs, Brinkmann and colleagues paved the way for this field in research by characterising the unique morphological properties of NETs. NETs are composed of an intricate network of sticky double-stranded DNA (dsDNA) coalesced with proteins, such as histones and granules, that are designed to neutralise foreign pathogens and elicit an inflammatory response. NETs were initially unveiled as an immunological defence mechanism against bacteria, with neutrophils releasing NETs in response to both Gram-positive and Gram-negative bacteria, including *Staphylococcus aureus* (*S. aureus*), *Salmonella typhimurium*, and *Shigella flexneri* [8]. Later research revealed that NET-based immunity extended to other types of pathogens, including viruses (i.e., *human immunodeficiency virus 1*), fungi (i.e., *Candida albicans* (*C. albicans*)), and parasites (*Leishmania amazonensis*) [9,10,11].

## 2. NETs in Immunity and Pathology

NETs provided a unique form of defence against pathogens by assembling adherent DNA into web-like structures, allowing them to physically trap pathogens and prevent their dissemination. This is particularly useful in targeting pathogens that clump together (i.e., *Mycobacterium bovis*) or are too large to be phagocytosed (i.e., *C. albicans*), as NETs can restrict their growth and facilitate their elimination via the antibacterial constituents coalesced with the DNA [12,13]. These constituents include histones, neutrophil elastase (NE), and myeloperoxidase, proteinase 3 (PR3), and defensins [14] (Figure 2).

Although histones are primarily recognised for their role in DNA packaging, they have long been known to possess antibacterial properties [15,16]. For decades, the reason behind these bactericidal abilities remained unclear, given that histones are typically contained within the nucleus. It was only with the discovery of NETs that their extracellular role became clear. Expulsion of histones via NETs allows them to promote bacterial lysis by stabilising LL-37—a pore-forming peptide—and enhancing membrane permeation, resulting in leakage of bacterial components out of the cytoplasm [17]. Due to the toxicity of histones to the host, their NET-bound nature aids in preventing dissemination and off-target damage [18,19,20].

MPO is the largest and most abundant enzyme in neutrophils, accounting for ~5% of the dry weight of the cell [21]. Neutrophil activation causes the release of MPO from azurophilic granules into the phagosome or extracellular space. In the phagosome, an enclosed and acidic environment, MPO catalyses the production of HOCl, a potent antimicrobial oxidant, and its activity level is enhanced under this acidic environment [22]. In NETs, however, the catalytic activity of MPO is comparatively reduced, given that the extracellular pH is typically higher (e.g., ~7.4 in blood). 

Similarly, NE is released from azurophilic granules into phagosomes, where it aids in bactericide activity by degrading virulence factors on bacterial surfaces. For instance, pneumococcal surface protein A—a protein required for bacterial adhesion to host cells and immune evasion—is cleaved by NE, thereby reducing the virulence of *S. pneumoniae* [23]. Proteinase 3—another azurophilic serine protease—is an enzyme that kills bacteria in a similar way to NE by degrading outer membrane proteins on bacterial surfaces, such as flagellin [24]. In NETs, NE and PR3 remain active for a longer period as they are embedded in DNA, making them more difficult to access for neutralisation by protease inhibitors like α1-antitrypsin.

Research into NETs gained increasing attention, primarily because they have been recognised to play a significant role in numerous diseases. NETs have been associated with autoimmunity, including diseases such as systemic lupus erythematosus (SLE), rheumatoid arthritis (RA), type I diabetes, and several forms of small vessel vasculitis (SVV). Their prevalence in autoimmunity can largely be attributed to their coalesced proteins, which can act as a rich source of autoantigens. For example, in a subtype of SVV called anti-neutrophil cytoplasmic antibody (ANCA)-associated vasculitis (AAV), MPO and PR3 are the targets of autoantibodies. This interaction can lead to the activation of the complement system. C5a of the alternative complement pathway binds to its receptor, C5a receptor (C5aR), on neutrophils, thus triggering the release of NETs [25].

NETs can amplify the complement system by acting as a scaffold for complement components. Properdin, found in secondary granules, is released alongside NETs, leading to the recruitment of C3b and factor B, and ultimately results in the formation of C3 convertase [26,27]. NETs also provide a structural platform for alternative complement activation [27]. This leads to increased C5a levels, inducing greater neutrophil recruitment and NETosis, thereby propagating a vicious cycle of inflammation.

NETs are frequently associated with inflammatory diseases. In these cases, there is dysregulation in the activation of neutrophils in response to infection or injury. In severe acute respiratory syndrome coronavirus 2 (SARS-CoV2), activated platelets—a potent stimulator of neutrophils—become trapped in the pulmonary microcirculation where neutrophils are found, thus leading to an interaction between the two, which results in enhanced NET formation [28]. Additionally, the DNA backbone of NETs can act as a physical barrier, impairing the clearance of the proinflammatory mediators entangled with them.

In cancer research, numerous studies have highlighted the substantial role NETs play in promoting tumour progression. Within the tumour microenvironment, NETs can facilitate invasion into surrounding tissues and metastasis by degrading extracellular matrix components through proteins NE and matrix metalloproteinase-9 (MMP-9). NETs can also enhance tumour survival by forming a physical barrier against cytotoxic immune cells, thus impairing T cell-mediated tumour clearance.

Given the association between NETs and the pathogenesis of numerous diseases, NETs have become an attractive therapeutic target. Preclinical studies have found great success in inhibiting NETs with various drugs that target different proteins in the signalling pathways to NETs release; however, few of them make it to clinical trials, and none have been approved for clinical use. This review seeks to answer why, despite the enormous potential of NET inhibitors, NET-targeted therapies are failing to reach clinical approval. We will discuss the different approaches to targeting NETs, the completed and ongoing clinical trials on NET-based therapies, and the biggest challenges in translational therapy surrounding NETs. Understanding these challenges is essential for refining therapeutic approaches and ultimately improving clinical outcomes for patients affected by NET-driven diseases.

## 3. Mechanisms of NET Formation

It is important to comprehend the underlying mechanisms behind the various intracellular signalling pathways that lead to NETs, as their implication in autoimmune diseases, inflammatory disorders, and cancer is predicated on their dysregulation. A better understanding of these signalling pathways is crucial for identifying therapeutic targets.

There are numerous pathways by which NETs are formed, leading to the release of various types, often categorised into lytic, vital, and mitochondrial NETs. These pathways determine what signalling components become active, the fate of the neutrophil, and the morphology of the NETs produced. Thus, this review will cover various types of NET stimulants and the specific pathway they induce (Figure 3).

### 3.1. Types of NETs

#### 3.1.1. Lytic NETosis

As its name implies, lytic or suicidal NETosis involves the slow release of NETs that ultimately result in neutrophil death. This canonical pathway was first comprehensively characterised by Fuchs et al. in 2007, describing the key steps prior to the release of lytic NETs, which were later confirmed by future studies [29]. The formation of lytic NETs primarily involves the generation of reactive oxygen species (ROS) derived from NADPH oxidase 2 (NOX2). NOX2 initially generates superoxide anions, which then undergo further reactions that result in hydrogen peroxide, hydroxyl radicals, and hypochlorous acid, which are collectively known as ROS [30]. Production of ROS causes the release of NE and MPO from azurophilic granules and enhances the activation of NE by oxidising and deactivating alpha-1-antitrypsin, an endogenous inhibitor of NE. This allows NE to play its role in processing histones—thereby promoting chromatin decondensation—and actin degradation in preparation for membrane rupture and NET release. Additionally, the release of MPO enables its enzymatic function of converting hydrogen peroxide to hypochlorous acid, a potent antimicrobial oxidant released with NETs. MPO can colocalise to the nucleus with NE to further promote chromatin decondensation. However, the PAD4 enzyme, activated by calcium ions, also plays a major role by citrullinating histones bound to chromatin, thus loosening their structural support and promoting their decondensation [31,32]. Finally, nuclear and plasma membrane rupture occurs when NE cleaves gasdermin D in the cytosol, activating its N-terminus fragment, forming pores in the membranes for extrusion of chromatin into the cytosol, and NETs into the extracellular space [33,34].

#### 3.1.2. Vital NETs

In 2010, a new, rapid form of NET release was discovered in response to *S. aureus* stimulation [35]. Whilst previous studies have reported that neutrophils take over three hours to release NETs, Pilsczek and colleagues observed NET-like structures as early as five minutes after stimulation, which were abolished upon DNase treatment, confirming their NET nature [35,36]. This rapid form of NETs maintained the integrity of the neutrophil plasma membrane despite DNA extrusion, resulting in cellular husks—termed anucleated cytoplasts—which retain their ability to phagocytose and migrate via chemokine signalling [36,37]. Unlike lytic NETosis, which requires membrane rupture, DNA release associated with these “vital NETs” involves the packaging of decondensed chromatin into vesicles, which can fuse with the plasma membrane to extrude the traps whilst keeping the cell membrane intact [35].

#### 3.1.3. Mitochondrial NETs

The discovery of a distinct form of NET formation involving the release of mitochondrial DNA, as opposed to nuclear DNA, was first reported in 2009 [25]. This process was observed when neutrophils were primed with GM-CSF and subsequently stimulated with LPS or complement component C5a [25]. Although lytic and vital NETs can involve mitochondrial ROS to some extent, mitochondrial NET release uniquely depends on it as its primary driver. Further studies demonstrated that neutrophil stimulation with ribonucleoprotein immune complexes (RNP ICs) leads to mitochondrial ROS production, playing a pivotal role in this pathway [38]. Excessive mitochondrial ROS generation can lead to organelle damage, which may be what prompts the extrusion of both mitochondrial ROS and DNA as a protective mechanism to prevent further cell injury [39]. Normal density granulocytes typically exhibit high mitochondrial membrane potential, with mitochondria evenly distributed throughout the cytoplasm. However, RNP IC stimulation was shown to reduce membrane potential, causing mitochondria to become hypopolarised and reposed along the lines of the plasma membrane. This is likely the mechanism by which mitochondrial NETs are formed and released.

### 3.2. Signalling Pathways of NET Formation by Stimulants

#### 3.2.1. Phorbol 12-Myristate 13-Acetate (PMA) Induction

PMA is an artificial stimulus that is commonly used to activate neutrophils and induce NETosis. This stimulant initiates the classical NOX2-dependent NETosis that involves the release of ROS, activation of NE and MPO, decondensation of chromatin, and rupture of the plasma membrane [40]. PMA directly activates protein kinase C (PKC), which, in turn, activates NOX2 to generate ROS, leading to the signalling cascade that results in membrane rupture and lytic NETs [41].

#### 3.2.2. Lipopolysaccharide (LPS) Induction

Bacterial LPS stimulates the NET formation pathway through recognition by toll-like receptor 4 (TLR4). Activation of TLR4 initiates a downstream signalling cascade that is mediated by c-Jun-N-terminal kinase (JNK) [42]. This was demonstrated when treatment of human neutrophils with SP600125 and TCSJNK6o—inhibitors of JNK—was able to reduce NETs induced by LPS [42]. Given that JNK functions upstream of NOX2, and that NOX2 ROS is required for LPS-induced NETs—as shown by NET inhibition following diphenyleneiodonium chloride (DPI) treatment, an inhibitor of NOX2-derived ROS—JNK activation appears essential for the production of ROS by NOX2 [42]. ROS activates key molecules in the pathway, such as NE, MPO, and PAD4, leading to the expulsion of NETs.

#### 3.2.3. Complement Induction

The alternative complement system plays a key role in driving various forms of NET release, including lytic and vital NETs [25]. A central component of this pathway involves C5a, a neutrophil chemoattractant and activator, which can bind to C5a receptors (C5aR) on neutrophils and trigger a cascade of intracellular signalling events. One important consequence of this interaction is the mobilisation of intracellular calcium stores, particularly the endoplasmic reticulum, as well as extracellular calcium influx through store-operated calcium channels [43]. Increased levels of intracellular calcium activate PAD4, thereby promoting citrullination of histones, leading to chromatin decondensation and subsequent NET formation [31].

It has been shown that C5a-C5aR signalling does not engage the spleen tyrosine kinase (Syk) pathway, with Syk-deficient neutrophils providing normal responses to C5a stimulation [44]. However, activation of the Akt, ERK, and MAPK pathways appears to proceed normally despite the lack of Syk kinase activation.

#### 3.2.4. Calcium Induction

Ionomycin and A23187 (calcimycin) are polyether antibiotics that are commonly used for NET induction as calcium ionophores. They increase intracellular calcium levels in neutrophils by facilitating the influx of extracellular calcium ions across the plasma membrane and mobilising calcium from intracellular stores, such as the endoplasmic reticulum [45,46]. This sharp increase in intracellular calcium levels directly activates PAD4, driving histone modification and chromatin decondensation. Simultaneously, excessive calcium accumulation in the mitochondrial matrix induces the opening of the mitochondrial permeability transition pore (mPTP), disrupting the normal functioning of the electron transport chain and resulting in increased production of mitochondrial ROS [31,47]. Unlike the classical NOX2-dependent NETosis, this calcium-driven NET formation occurs independently of NOX2-derived ROS and instead relies on mitochondrial ROS as the primary oxidative driver of chromatin decondensation and NET release.

#### 3.2.5. Antigen-Antibody Complex Induction

Fc gamma receptor IIa (FcγRIIa) expressed on neutrophils recognises IgG antibody-antigen complexes [48]. This leads to the activation of the tyrosine kinase Syk and the downstream signalling pathway involving the recruitment of phosphoinositide 3-kinase (PI3K), which then phosphorylates Akt. The mechanisms of this NET formation pathway beyond this step are ambiguous; however, it is likely that phosphorylation of the regulatory subunit p47phox is conducted by Akt, with studies showing that activation of this subunit is necessary for NOX2 activation [49,50,51]. This leads to NOX2-derived ROS production and ultimately results in NOX2-dependent, PAD4-dependent lytic NETosis [52].

#### 3.2.6. Osteopontin Induction

Osteopontin (OPN) is a calcium-binding phosphoglycoprotein that is involved in neutrophil migration and adhesion [53]. It binds to the α9β1 integrin expressed on the neutrophil cell surface, activating a downstream signalling cascade that involves a NOX2-dependent PI3K/Akt pathway [54]. NOX2-derived ROS leads to PAD4 activation, and subsequent chromatin decondensation and NET release [54].

#### 3.2.7. Hepoxilin A3 Induction

Hepoxilin A3 (HXA3) is a bioactive lipid derived from arachidonic acid that is produced by epithelial cells in response to *P. aeruginosa* [55]. HXA3 acts as a potent chemoattractant that facilitates the movement of neutrophils across epithelial barriers; however, the exact receptors that HXA3 binds to are not fully elucidated. HXA3 has been shown to activate neutrophils and induce an increase in intracellular calcium ions in neutrophils [56]. Since calcium influx is critical for many NET formation pathways, the capacity of HXA3 as a direct inhibitor of NETs was tested [57]. Interestingly, HXA3 engaged different types of NET formation pathways depending on the dose used for stimulation. A higher dose (10 µg/mL) elicited a NOX2-independent NET release and was unaffected by DPI. In contrast, a lower dose (2.5 and 5 µg/mL) induced a NOX2-dependent NET response, with DPI significantly reducing NET release.

## 4. Biochemical Modifications to NETs

NETs, or NET components, can undergo various biochemical modifications that can affect their persistence and stability in tissues, as well as their immunogenicity.

NET DNA has a high content of 8-OHG, a marker of oxidative damage, likely caused by ROS generated during NET formation [58]. The expulsion of high concentrations of oxidised DNA via NETs in the extracellular space has been found to elicit a type I interferon (IFN) response by acting as a damage-associated molecular pattern (DAMP) for plasmacytoid dendritic cells (pDCs) and is particularly relevant in SLE [58,59]. Furthermore, oxidised DNA appears to be resistant to degradation by Three Prime Repair Exonuclease 1 (TREX1), which is known to mediate clearance of NETs, along with DNase I and II [58,60]. This modification is also associated with ageing, as NETs in elderly patients exhibit higher levels of oxidation and have higher resistance to clearance by DNase I [61].

In addition to the citrullination of histones, other post-translational modifications, such as acetylation and methylation, have been identified on NETs. Similar to citrullination, the acetylation of histones leads to chromatin loosening and decondensation, while methylation is more commonly associated with transcriptional repression. In SLE, these modifications have been linked to an increased immunostimulatory capacity of NETs, enhancing their ability to activate macrophages [62,63].

Although NETs are increasingly recognised for their importance in diabetes-related chronic inflammation, the role of glycation in modifying NETs remains poorly understood. Advanced glycation end products (AGEs) form readily in glucose-dense environments and are typical of diabetes. AGEs are known to alter proteins’ structure and function. As NETs are rich in glycation-prone residues, such as lysine and arginine, their glycation could affect their susceptibility to degradation, as well as their proinflammatory properties. The abundant histones in NETs are especially susceptible to glycation due to their high content of these amino acids.

## 5. NET-Targeted Therapies

Given the widespread involvement of NETs in a myriad of diseases, there is a growing urgency to develop effective therapeutic strategies that mitigate their pathological effects. While the development of novel drugs is important, repurposing existing immunomodulatory drugs as NET inhibitors provides a rapid way to drive therapeutics from the lab into the clinic with their well-established safety and tolerability data profiles. In this review, therapeutic approaches are ordered based on the stage of NET formation they target and categorised according to preclinical (Table 1) and clinical studies (Table 2), as this can help to distinguish whether they address the underlying disease mechanism or alleviate symptoms (Table 1). This distinction is vital when identifying gaps in current treatment regimens and optimising disease-specific interventions.

### 5.1. Repurposing for NET Inhibition

Therapeutic strategies aimed at mitigating the harmful effects of NETs focus primarily on preventing their formation, but can also target their harmful components or enhance their clearance. For instance, early-stage inhibitors focus on blocking the inducers of NET formation, thereby blocking its initiation, but could also target ROS production or activity, which occurs relatively early in most pathways. Late-stage inhibitors aim to block signalling events that are further downstream, including PAD4 activation and membrane rupture, as well as neutralise NET components that cause damage, such as histones. Therapies that enhance NET clearance focus on accelerating the removal of NETs, thereby resolving the inflammation caused by these structures.

#### 5.1.1. Early-Stage Inhibitors

##### Eculizumab

Eculizumab, sold under the brand name Soliris, is a recombinant monoclonal antibody that targets C5 and prevents its cleavage to C5a and C5b. It is currently approved for the treatment of paroxysmal nocturnal haemoglobinuria (PNH), a blood disorder that involves premature haemolysis and haemoglobinuria. The recent FDA approval of Avacopan, a C5aR inhibitor, for clinical treatment of AAV—an autoimmune disease where NETs are a major driver of pathology—shows promise of therapeutic C5 inhibition in NET-mediated diseases. As C5a is a potent inducer of NETs, eculizumab has been explored for its role in NET formation and has been implicated in inhibiting NETs in the context of PNH, as it was able to reduce markers of NET formation in vivo [64].

Although other C5 inhibitors exist, such as iptacopan, eculizumab shows greater potential due to its established safety profile and proven ability to reduce C5a-driven inflammation [80]. In contrast, iptacopan has limited evidence regarding its effects on NET formation, as it has not been directly studied in the context of NETs.

##### Tocilizumab

Tocilizumab, commonly sold under than brand name Actemra, is a recombinant monoclonal antibody that binds to IL-6 receptors, preventing its interaction with IL-6 cytokines, thereby inhibiting its proinflammatory signalling pathway. 

The IL-6 cytokine has been shown to directly induce NETs comparable to LPS, after 16 h of stimulation [81]. Although the exact mechanism behind this phenomenon remains a mystery, it was recently found that tocilizumab treatment reduced the NET markers dsDNA, MPO-DNA, and citrullinated histone 3 in the serum of patients with an ST-segment elevation myocardial infarction compared to placebo controls [73]. The clinical trial this was based on—the ASSessing the Effect of Anti-IL-6 Treatment in Myocardial Infarction (ASSAIL-MI) trial—reported no significant difference in serious adverse events in the treated group compared to the placebo. It is worth noting that tocilizumab is currently FDA-approved for RA and giant cell arteritis, and is generally considered safe, although it can increase the risk of infections typical of immunosuppressive treatments [82,83]. Further experiments should be conducted to determine the efficacy of tocilizumab as a NET inhibitor in vivo.

##### Metformin

Metformin is a small-molecule drug that inhibits the mitochondrial electron complex 1, thereby reducing the formation of superoxides and other ROS derivatives. Inhibition of complex 1 results in reduced ATP production, thereby activating AMP-activated protein kinase—a sensor that detects low levels of ATP—leading to a decrease in hepatic gluconeogenesis [84]. Thus, metformin has been used for the treatment of type II diabetes since the 1950s.

More recently, it has been investigated for NET-related diseases as most pathways to NETosis involve the utilisation of either NOX2- or mitochondrial-derived ROS, or both. In 2018, the effects of metformin on circulating NETs biomarkers of patients with pre-diabetes were compared to a placebo group control, with metformin eliciting a significant reduction of dsDNA, NE, PR3, and histones in the plasma after two months of therapy [74]. This was supported by the observation that metformin reduced ionomycin-induced NET formation in human neutrophils in vitro.

NETs contribute negatively to COVID-19 severity, with elevated levels of neutrophils found in the lungs, particularly in the pulmonary circulation, where they interact with activated platelets to trigger NET formation [28]. The release of NET components, including DNA, MPO, and citrullinated histones, exacerbates lung inflammation and has been shown to correlate with COVID-19 severity [85,86]. Indeed, the difference between symptomatic and asymptomatic COVID-19 patients is that persistent elevations in NETs post-diagnosis are seen in symptomatic but not asymptomatic patients. Promising NET inhibitor drugs, such as metformin, have therefore been recently explored in the context of COVID-19.

Conflicting findings have been reported regarding metformin’s effectiveness in COVID-19 patients, particularly between observational studies and the TOGETHER trial [87,88]. This randomised clinical trial included 418 SARS-CoV-2 patients who were randomly assigned to receive either metformin (750 mg) or a placebo twice daily for 10 days [87]. A meta-analysis of five observational studies (total *n* = 8121) found that metformin significantly reduced COVID-19 severity and mortality [88]. However, the TOGETHER trial found no significant difference in clinical improvement or mortality between the metformin and placebo groups. This discrepancy may be due to differences in the timing of metformin administration. In the observational studies, metformin was taken before COVID-19 diagnosis, potentially offering protective effects. In contrast, the TOGETHER trial administered metformin after diagnosis and symptom onset, at which point, significant NET formation had already occurred. By this stage, metformin could no longer alter disease progression, as impaired NET degradation had become a contributing factor. Prophylactic use of metformin may be more effective in inhibiting NET formation during the early stages of the disease, when NETs play a key role in its pathophysiology.

NETs also play a significant role in the pathogenesis of SLE by exposing nuclear autoantigens that drive autoantibody production [89]. These NET-associated components stimulate plasmacytoid dendritic cells (pDCs) to secrete IFNα, a key inflammatory mediator in SLE [90]. Mitochondrial DNA within NETs has been identified as a potent inducer of IFNα through activation of TLR9 on pDCs [91]. A proof-of-concept randomised clinical trial investigated metformin’s ability to inhibit NET formation in the context of SLE [67]. In vitro, metformin treatment reduced mitochondrial DNA-induced IFNα production by pDCs. In a clinical setting, metformin was tested as an add-on therapy to standard SLE treatment, leading to reduced clinical flares, decreased prednisone use, and a lower body mass index. Additionally, this clinical trial saw that metformin was generally well-tolerated; however, mild gastrointestinal discomfort was experienced in 5 out of 56 patients.

##### Fostamatinib

Fostamatinib is a prodrug that targets the Syk tyrosine kinase, blocking the signalling pathways downstream of this. It is currently approved for use in chronic immune thrombocytopenia—a blood disorder characterised by platelet destruction. As Syk is an important kinase in various NET formation pathways, such as FcγRIIA-associated NETs, its potential as a NET inhibitor was explored. NETs induced by COVID-19 patient plasma were reduced upon fostamatinib treatment of healthy human neutrophils in a concentration-dependent manner [65]. Indeed, fostamatinib has undergone phase 3 clinical trials involving patients with COVID-19 and hypoxemia [92]. In this clinical trial, 400 patients were randomised to receive fostamatinib (150 mg) via oral administration or a matching placebo twice daily for 2 weeks as an add-on to current standard care. Given that COVID-19 is a disease that causes lung injury, the primary outcome was oxygen-free days as a result of the duration of supplemental oxygen use. The mean number of oxygen-free days between the treatment and placebo groups was not significantly different, and neither was mortality. Furthermore, the proportion of patients who experienced concerningly high levels of aspartate aminotransferase (>128 U/L in men and >104 U/L in women) was higher in the fostamatinib group compared to the placebo.

##### SkQ1

SkQ1, also known as visomitin, is a small-molecule antioxidant drug that localises to the mitochondrial membrane and scavenges excess mitochondrial ROS. Developed in Russia, originally for anti-ageing therapies, it is now being explored in cancer therapy, neurodegenerative diseases, and NET modulation. Calcium ionophore A23187-induced NET release in human neutrophils involves both NOX2- and mitochondrial-derived ROS [46]. Treatment with SkQ1 inhibited mitochondrial ROS production and saw a reduction in A23187-induced NETs [46]. Whilst there are no clinical trials looking into the NET-inhibiting effects of SkQ1, it has undergone up to phase 3 clinical trials for the treatment of dry eye disease (*n* = 444), from which its safety profile can be extrapolated [93,94]. High (1.55 µg/mL) and low (0.155 µg/mL) doses of SkQ1 ophthalmic solution were reportedly safe and well-tolerated relative to the placebo, with no serious adverse events that occurred, and no subjects who discontinued due to side effects [93]. As a result, there are plans to proceed with further clinical trials en route to its approval for clinical use in the United States, though it is worth noting that this drug has been approved for dry eye clinical use in Russia since 2011. There is potential for the repurposing of SkQ1 towards the treatment of NET-based diseases, although further preclinical studies in vivo should be undertaken to determine its efficacy in NET inhibition.

#### 5.1.2. Late-Stage Inhibitors

##### Colchicine

Colchicine is an ancient drug that is derived from the autumn crocus plant, with records showing its use for gout-like symptoms by Ancient Egyptians and Greeks. Now, we know this to be an agent that disrupts the assembly of microtubules by inhibiting β-tubulin polymerisation. Fixation of the cytoskeletal structure in primed neutrophils prevents chromatin swelling and, ultimately, NET release. The inhibitory capacity of colchicine on NETs has recently been demonstrated in experimental autoimmune myositis (EAM) and acute myocardial infarction models of mice [66,95]. Colchicine appears to be comparable to Cl-amidine—a pan-PAD inhibitor and a well-known inhibitor of NETs—in its capacity to reduce NET release both in vivo in EAM mice and in vitro using human neutrophils [95,96]. In addition, a randomised clinical trial, looking at the effects of colchicine as a therapeutic strategy against moderate to severe COVID-19, has found that it improved clinical outcomes, reducing the supplemental oxygen and hospitalisation time required by patients with COVID-19 [97]. Although NET release was not directly measured in this study, the high inflammatory status of COVID-19 is characterised by high levels of IL-1β, IL-6, IL-18, and TNF, all of which are known to induce NETs [81,98,99,100]. Given its extensive history for medicinal use, colchicine is generally well-tolerated with well-known adverse effects of gastrointestinal symptoms, including diarrhoea, nausea, and abdominal pain [97].

##### BB-Cl-Amidine

BB-Cl-Amidine is a pan-PAD inhibitor and was developed by Knight et al. (2015) specifically for the purpose of developing a PAD inhibitor with improved cellular bioavailability compared to Cl-Amidine [67]. It irreversibly binds to PAD enzymes, thereby inhibiting their ability to deiminase histones, a process that is required for NET release. BB-Cl-Amidine is more readily taken up by cells and has a longer in vivo half-life than Cl-Amidine (1.75 h vs. 15 min). BB-Cl-Amidine was first used as a therapeutic strategy against SLE, a condition in which there are elevated levels of NETs and impaired clearance, leading to autoantigen exposure, type I IFN activation, and endothelial damage [101]. Treatment with BB-Cl-Amidine resulted in a significant reduction in NET formation by neutrophils of lupus-prone mice (without impeding ROS production by NOX2), which is crucial in many essential pathways for maintaining homeostasis [67]. This led to improvements in vascular function by increasing endothelium-dependent vasorelaxation in the lupus-prone mice, attenuating a symptom of SLE.

Whilst BB-Cl-Amidine has shown promise in preclinical murine models regarding its efficacy for NET inhibition, its safety profile is not well understood and would have to be ascertained before proceeding to clinical trials.

##### GSK484

A more targeted drug relative to BB-Cl-Amidine is GSK484, a small-molecule inhibitor that is specific against PAD4. Similar to BB-Cl-Amidine, there are currently no clinical trials in which GSK484 is used. However, various preclinical studies have found that GSK484 is effective in the reduction of NETs, making it a promising therapeutic strategy for NET-associated diseases. The first time the role of PAD4 in NET formation was elucidated was in 2015 using GSK484 [68]. The authors found that pretreatment of mouse neutrophils with this drug diminished ionomycin-induced citrullination of histones and NET formation in a dose-dependent manner, with no sign of chromatin decondensation and nuclear swelling.

GSK484 has also been investigated for cancer therapy. In liver metastases from colorectal cancer (CRC), PAD4 and citrullination of histones are upregulated and associated with poor prognosis in CRC patients [102,103]. NETs aid in enhancing tumour metastasis by facilitating tumour cell adhesion to endothelial cells and potentially by promoting extravasation [104,105]. Wang et al. (2023) showed that treatment of CRC cell lines (SW480 and HCT116) with GSK484 improved their radiosensitivity and reduced A23187-induced NET formation in vitro [103]. GSK484 also incurred significant NET reduction in both excised tumours and plasma of mouse xenograft tumour assay models.

Unlike BB-Cl-amidine, GSK484 is reversible and highly specific, only binding to and inhibiting PAD4. Against a panel of 50 unrelated proteins, GSK484 showed negligible off-target binding [68]. It is also important to note that PAD4-mediated NET formation is not essential for immunity against all infections. Hemmers and colleagues (2011) demonstrated that PAD4-deficient mice were not more susceptible to influenza A infection compared to wild-type mice, suggesting that inhibition of this type of NET formation using GSK484 will not compromise host defence against this virus [106]. Indeed, there are other pathways leading to NET formation that occur independent of PAD4. Guiducci et al. (2018) found no significant difference in NET formation between PAD4-deficient mice and wild-type mice subsequent to infection with *C. albicans* [107]. This further supports the potential of PAD4 inhibitors as viable therapeutic strategies, as their effects on NET inhibition are selective for certain disease contexts without broadly impairing immune function.

##### BAY 85-8501

BAY 85-8501, also known as brensocatib, is a fifth-generation selective inhibitor of human NE derived from the fourth-generation BAY-678. BAY 85-8501 has primarily been explored for its potential therapeutic effects in pulmonary diseases and has undergone a phase 2 clinical trial for the treatment of non-cystic fibrosis bronchiectasis [78]. This disease is characterised by chronic inflammation of the lungs, with increased disease severity being associated with the inflammatory mediator NE. NE levels in the sputum can be indicative of disease progression and are therefore a potential therapeutic target. The clinical trial lasted 28 days and included patients who received either BAY 85-8501 (1 mg) or a placebo once daily. Although they found that BAY 85-8501 did not improve pulmonary function or patient quality of life, its safety and tolerability in patients were generally favourable, with no clinical difference in outcome relative to the placebo. Of note, one patient discontinued due to mild vomiting that was caused by BAY 85-8501.

Although there are no publicly available studies that have looked at BAY 85-8501 specifically as a NET inhibitor therapy, it has previously been shown that NE is essential for lytic NETosis. This was first seen in 2010 when Papayannopoulos et al. described the role of NE in histone modification and chromatin decondensation, and later, reinforced in 2014, by using NE-deficient Elane-KO mice, which showed impaired NET formation [108,109]. In addition to BAY 85-8501 being administered orally, making it a more convenient treatment for patients, the efficacy of this drug in preclinical research and safety in clinical trials makes it a promising candidate for targeting NETs in diseases such as SLE and ANCA-vasculitis, where they play a pathological role.

##### AZD9668

AZD9668, also known as Alvelestat, is a third-generation reversible inhibitor of human NE that has undergone clinical trials for pulmonary diseases, including bronchiectasis, cystic fibrosis, and chronic obstructive pulmonary disease (COPD) [75,76,77]. In a phase 2 clinical trial for bronchiectasis, 38 patients were randomised to receive either AZD9668 (60 mg) or a placebo twice daily for 4 weeks [75]. Similarly, a trial for cystic fibrosis included 56 patients who were administered AZD9668 (60 mg) twice daily for 4 weeks [76]. A COPD clinical trial involving a large sample size of 838 patients who were assigned AZD9668 at varying doses of 5, 20, or 60 mg or a matching placebo twice daily for 21 weeks [77]. Despite being well-tolerated with no notable adverse events reported, AZD9668 did not lead to significant improvements in clinical outcomes or patient quality of life across all these trials.

Nevertheless, its reversible nature reduces the risk of toxicity, making it an appealing candidate for repurposing once its efficacy in NET inhibition is established. Compared to the fourth-generation NE inhibitor BAY-678, AZD9668 may be more promising due to its oral bioavailability, which is particularly crucial for chronic conditions. BAY-678 lacks oral bioavailability, limiting its practicality as a long-term treatment option. Furthermore, unlike BAY 85-8501 and AZD9668, BAY-678 has yet to advance to phase 2 clinical trials, and its potential effects on NETs remain unexplored.

##### CIT-013

CIT-013 is a monoclonal antibody specific against citrullinated histones H2A and H4, key markers of NET formation. By binding to these modified histones, CIT-013 effectively inhibits NET release, as has been demonstrated in mouse neutrophils activated with disease-relevant stimulants, including bleomycin-induced pulmonary fibrosis, dextran sulphate sodium-induced colitis, and LPS-induced sepsis, all of which are pathologically mediated by PAD4-dependent NETs [69]. In all tested models, CIT-013 treatment significantly reduced disease severity, highlighting its ability to mitigate NET-driven inflammation and tissue damage. Notably, while NET formation was successfully inhibited, neutrophil recruitment remained uninfluenced, suggesting that it selectively targets NET-associated pathology without compromising essential neutrophil functions.

CIT-013 has undergone a phase I clinical trial, evaluating its efficacy in humans in vivo, as well as its safety profile and tolerability in 73 healthy volunteers [110]. Single ascending doses (0.1 to 3 mg/kg) of intravenous CIT-013 were administered, and found that a dose of up to 0.3 mg/kg was well-tolerated, with higher doses causing chest discomfort and elevated inflammatory markers. Pretreatment with 0.3 or 0.9 mg/kg of CIT-013 significantly reduced LPS-induced systemic NET formation relative to a placebo. Subcutaneous administration was also evaluated, showing a bioavailability of 66% compared to intravenous administration, and was generally well-tolerated. CIT-013 is planned to undergo phase 2 clinical trials involving patients with RA.

##### Disulfiram

Disulfiram, an FDA-approved treatment for chronic alcohol dependence, has been in use since the late 1940s. More recently, it was identified as an inhibitor of NET formation, specifically blocking PMA-reduced NET release in both mouse and human neutrophils in a dose-dependent manner [70]. This effect is most likely due to its ability to inhibit gasdermin D polymerisation and activation, which are essential for nuclear and cellular membrane rupture during NETosis [34,111]. In vivo, disulfiram reduced NET formation in COVID-19-infected golden hamsters, leading to improved lung histology [70]. Building upon this finding, a recent clinical trial investigated the safety and efficacy of disulfiram in patients with moderate COVID-19 [112]. This trial consisted of 140 participants who were to receive a daily dose of disulfiram (500 mg) or a placebo in addition to standard care and were assessed on days 8 and 15. Contrary to the findings in rodents, this trial found no clinical improvements in the COVID-19 patients. Additionally, adverse events were more common in the disulfiram-treated group (67.6%) compared to the placebo group (37.7%), which were primarily driven by lactate dehydrogenase elevation, D-dimer increase, and dyspnoea. However, there was no difference in serious adverse events between the treatment and placebo groups, with disulfiram not causing any major safety concerns.

As gasdermin D is also a crucial downstream enzyme in the inflammatory cell death pathway, pyroptosis, abrogation of pore formation prevents this pathway from concluding [111]. This makes disulfiram promising for repurposing, not only for NET-associated diseases, but also for counteracting conditions associated with chronic inflammation.

##### NINJ1 Monoclonal Antibodies

Ninjurin-1 (NINJ1) is a cell surface protein that plays a crucial role in plasma membrane rupture of lytic cell death pathways, such as apoptosis and pyroptosis, acting downstream of gasdermin D [113]. Given that a similar process occurs in lytic NETosis regarding plasma membrane rupture, NINJ1 came under investigation for its involvement in NET formation. A new study, exploring acute oxalate nephropathy—a condition involving the accumulation of calcium oxalate crystals in renal tissues, leading to functional impairments—highlighted NINJ1 as a key contributor in the formation of NETs [71]. Neutrophil infiltration of the kidneys, as well as NET release, increased upon oxalate treatment and disease progression [71,114]. Notably, NINJ1 deficiency in myeloid cells significantly reduced NET formation both in vitro and in vivo, supporting its role in this process.

Kayagaki and colleagues (2023) demonstrated that blocking NINJ1-mediated membrane rupture can have therapeutic potential [115]. They developed monoclonal antibodies that prevent NINJ1 oligomerisation, thereby attenuating plasma membrane rupture. Since NETs are known to exacerbate liver inflammation and injury in hepatitis, hepatitis mouse models were treated with these antibodies, resulting in improved liver morphology. Further, treatment with NINJ1 antibodies did not seem to alter apoptotic clearance by phagocytes, making it a promising candidate for NET inhibition therapy.

### 5.2. Enhancing NET Clearance—DNase

DNase I, commonly marketed as a recombinant form called Dornase Alfa, is an enzyme that degrades DNA. It is clinically approved for cystic fibrosis, in which it breaks down extracellular DNA present in mucus, thereby reducing its viscosity and improving lung clearance [116,117]. As NETs are primarily made up of DNA, it is common in research to use DNase I to ascertain the role of NETs in biological pathways and diseases. Notably, DNase I has been explored regarding its therapeutic potential in intestinal ischemia–reperfusion (I/R) injury [72]. The study used a rat model of I/R injury and found that NET levels were elevated in the ileum tissues during injury, which were significantly reduced upon DNase I intravenous (IV) injection. Additionally, treatment with DNase I reduced the typical early IL-6 and TNF response subsequent to I/R injury, as well as inhibited histopathological changes due to injury. DNase I has also been subjected to a recent clinical trial for the treatment of severe COVID-19 pneumoniae [79]. In this randomised trial, patients were treated with nebulised DNase I adjunct to standard treatments. This resulted in a reduction in inflammatory markers, leading to a reduced time to discharge over 35 days. These findings suggest that enhancing NET degradation using DNase I may be a promising strategy in limiting lung injury due to COVID-19. This may provide an indication that DNase I may also be of benefit in treating other NET-based diseases.

### 5.3. Other Potential Targets for NET Inhibition

#### 5.3.1. Pentraxin 3

Pentraxin 3 is a soluble pattern recognition receptor that plays an important role in innate immunity by sensing pathogens such as *S. aureus* and *Aspergillus fumigatus* [118,119]. Beyond pathogen recognition, pentraxin 3 is also responsible for detecting the histones and nucleic acids released from dead or dying cells to facilitate their clearance [120]. In sepsis, pentraxin has been shown to bind extracellular histones, mitigating their cytotoxic effects and protecting against histone-induced endothelial damage [18,121]. As NETs are a major source of extracellular histones, this suggests that pentraxin 3 contributes to the regulation of NET-associated damage by reducing its detrimental effects exerted by histones. This makes it a promising target for controlling NET-associated damage. Further studies are required to determine whether pentraxin 3 affects NET formation itself, as well as its potential role in mitigating NET-associated damage in other inflammatory and autoimmune diseases.

#### 5.3.2. Hepoxilin A3

As HXA3 is a potent inducer of NETs, it may be worthwhile investigating the effects of reducing its production on NET formation as a potential NET inhibitor therapy. Hepoxilins are generated through the 12S-lipoxygenase (12S-LOX) pathway of arachidonic acid metabolism [122]. Thus, using 12-LOX inhibitors such as baicalein and nordihydroguaiaretic acid could potentially inhibit NETs indirectly by reducing HXA3 generation [123,124].

## 6. Challenges in Translation

Despite the numerous successful NET inhibitors that have been found in preclinical studies, it begs the question as to why there have not been any NET inhibitors approved for clinical use. Given the significant involvement of NETs in the pathology of diverse diseases, from autoimmunity and cancer to thrombosis and sepsis, a successful NET inhibitor would potentially transform treatment paradigms across multiple fields.

### 6.1. Limitations of NET Inhibitor Laboratory Experiments

#### 6.1.1. Animal Models

One of the primary issues with translating treatments from a research setting to a clinical environment is the limitations of using animal models of disease. For example, research in AAV—an autoimmune disease characterised by autoantibodies against proteins on neutrophils and NETs—commonly uses an anti-myeloperoxidase IgG-induced vasculitis mouse model [125]. Whilst humans typically develop AAV gradually, this model requires the disease to be induced, causing acute inflammation that does not mimic the long-term dynamics of the inflammation seen in patients. This can result in an overestimation of the effectiveness of NET inhibitors, as their efficacy is observed only in the short term. This issue extends to animal models of other diseases. In atherosclerosis, commonly used animal models, such as LDL receptor-KO mice and apolipoprotein E-KO mice, take weeks or months to develop, whilst human atherosclerosis typically progresses over decades [126,127,128]. Similarly, rheumatoid arthritis models like K/BxN mice develop the disease within weeks or months, whereas human arthritis gradually progresses over years [129,130]. As a result, these animal models cannot accurately represent the mid- and long-term effects of using NET inhibitors in these diseases.

An obvious limitation of using mouse models for developing pharmacological drugs for humans is that mouse neutrophils vary greatly compared to human neutrophils. In humans, neutrophils are the most abundant type of leukocyte, making up ~50% of all leukocytes, whilst mouse neutrophils make up only ~20%, causing NET-driven diseases to be less severe in mice [131]. Mouse neutrophils also have a much lower propensity to form NETs compared to human neutrophils, which can NET even without strong stimulation, with ~80% of neutrophils undergoing NETosis within 3–4 h of *C. albicans* stimulation, whilst only ~30% of mouse neutrophils NET over a longer period of 16 h post-stimulation [132]. These factors would downplay what might otherwise be a greater effect of NET inhibitors on human disease when tested in mouse models. There are also morphological differences between human and mouse NETs, as they tend to be more compact in mice, with human NETs being more expansive and web-like in structure [132].

There are a few viable strategies to address these limitations. One approach is to use humanised mouse models by engrafting mice with human immune cells or genetically modifying them to express human receptors and proteins. Continuing with the example of the AAV mouse model, where many current models rely on anti-MPO induction of vasculitis, an improvement would be to generate mice expressing human MPO or PR3, the primary targets of ANCA, to better mimic the progression of human disease. Similarly, in atherosclerosis research, mice expressing human apolipoprotein variants could improve the accuracy of findings. The use of NSG mice could also be considered, as they lack mature T, B, and NK cells, and could therefore be reconstituted with human CD34 cells, making their immune cells human.

#### 6.1.2. In Vitro Experiments

A significant challenge that comes with neutrophil research is their sensitivity. Neutrophils are highly sensitive to handling and general mechanical forces, making their isolation difficult to perform without inducing artificial NET release [133,134]. Additionally, the short lifespan of neutrophils in vitro, which is typically no more than a few hours, makes it challenging to assess the long-term effects of NET inhibitors [135,136].

In many of the preclinical studies that reported success in using NET inhibitors, artificial stimulants (i.e., PMA, ionophores, and isolated pathogens) were used to trigger NET formation. Although they reliably induce NETs, it is through a method that does not accurately reflect the disease-specific triggers seen in vivo. For instance, PMA, one of the most used stimulants of NETs, activates PKC and initiates the NOX2-dependent pathway to lytic NETosis [40]. However, there are numerous pathological triggers that engage different pathways, such as bacterial sepsis through TLRs, and AAV through alternative complement activation [36,137]. Similarly, calcium ionophores such as A23187 and ionomycin (also used commonly in these preclinical studies) activate NETosis in a NOX2-ROS-independent pathway, which is uncommon in many diseases where NOX2 plays an important role [138,139]. Furthermore, these artificial stimulants induce NETs at higher rates and intensities compared to in vivo disease settings and are sometimes used in very high doses, which activate pathways that are typically not engaged [140].

This would overestimate the efficacy of NET inhibition by the study drugs. Finally, the administration of artificial stimulants directly to neutrophils neglects the interaction present in diseases such as thrombosis and sepsis, during which platelets bind to neutrophils to enhance NET release [141,142].

### 6.2. NETs Are a Biological Conundrum

The current literature about the biological mechanisms of NETs is likened to a bundle of convolutedly intertwined strings of incomplete information that is not easily untangled. This is in large part due to the heterogeneity of NETs, which poses a multitude of challenges when it comes to developing effective therapeutic inhibitors. One of the ways in which this heterogeneity manifests is through the numerous distinct pathways of NET formation, which result in different types of NETs being released. This means that the triggers, kinetics, and role/s of NETs can vary depending on the disease, making it difficult to develop a NET inhibitor that targets the specific NET type that is relevant and one that is effective across multiple conditions.

There is also the matter of timing the optimal stage during which NETs should be inhibited. Depending on the disease, NETs may be more important to inhibit at certain stages, and failure to treat NETs during this window will result in lacklustre outcomes. For example, NETs are extremely important in the very early stages of COVID-19. Hence, prophylactic treatment with NET inhibitors has shown clinical benefits [88]. However, NET-inhibiting treatments post-diagnosis are too late, as seen in the TOGETHER clinical trial, as the early release of NETs has already firmly established the disease [87]. In some cases, mistiming the administration of NET inhibitors may even result in exacerbated pathology. In sepsis, NETs are beneficial in the early stages of the disease as they aid in neutralising pathogens, whilst in the later stages, excessive NETs accumulate and cause hyperinflammation, which can lead to endothelial damage [143]. Therefore, inhibition of NETs during late-phase sepsis could prove beneficial, while early-stage inhibition has been shown to increase bacterial load [143].

This alludes to the role of NETs in host defence and the challenges of balancing the inhibition of NETs and suppressing immunity. Over the years, there have been a few infections for which NETs play a dominant role. These include bacterial infections (i.e., *K. pneumoniae* and *S. aureus*), which have evolved to evade phagocytosis, and fungal infections (i.e., *C. albicans*), which are too large to phagocytose [12,13]. Immunosuppression of NETs can increase patient susceptibility to these types of infections, in addition to combating pathogens. Indeed, D’Cruz and colleagues have found that administration of DNase I to IFN-γ-primed human and mouse neutrophils infected with methicillin-resistant *Staphylococcus aureus* (MRSA) resulted in a significant increase in bacterial growth compared to the control [144].

NETs also play a role in enhancing other innate immune cells and modulating the adaptive immune system. NETs promote macrophage polarisation to the proinflammatory M1 phenotype, as well as recruit more neutrophils and monocytes to sites of inflammation via IL-8 [145,146]. Additionally, NETs can prime CD4+ T cells and lower their activation threshold, thus adding another layer of consideration when designing NET inhibitors [147]. These factors make it difficult to balance the inhibition of NETs and their pathology in diseases, with their immunosuppression affecting both innate and adaptive immunity.

### 6.3. Bioavailability

Bioavailability is an issue when it comes to the development of any drug, and, unfortunately, DNase I presents multiple limitations in this aspect. DNase I has low stability in serum and is easily deactivated by heat and other environmental stimuli [148,149,150]. One major factor affecting its activity is G-actin, one of the most abundant proteins in most eukaryotic cells. G-actin binds to DNase I with high affinity, sterically inhibiting DNase I from catalysing the cleavage of phosphodiester linkages in DNA and performing its enzymatic role [151]. G-actin can be released into the bloodstream during cell injury and is the reason why it is not possible to administer DNase I systemically, given its short half-life in circulation [152]. It can also be toxic at high doses and may result in excessive cell death and tissue injury.

An alternative nuclease is DNase γ, also known as DNase I-like 3, and has long been recognised for its role in chromatin degradation during apoptosis. However, recent studies have demonstrated that DNase γ can also degrade NETs, as well as prevent NET-mediated vascular occlusion in patients with severe bacterial infections [153,154]. Its ability to effectively break down NET structures suggests similar therapeutic applications to DNase I, with an added benefit of increased stability in circulation, as it is not inhibited by G-actin [155]. The lack of clinical trials using DNase γ warrants further investigation of its safety profile and tolerability.

Emerging research is exploring the use of adeno-associated viruses (AAVs) in combination with DNase I as a therapeutic strategy against NET-mediated diseases. These viral vectors are commonly used in gene therapy to deliver therapeutic genes directly into target cells [156]. This approach could help overcome the issue of bioavailability by enabling cells to continuously produce DNase I, thereby reducing the need for frequent administration. Xia and colleagues investigated AAV-mediated DNase I delivery in CRC and found a significant reduction in liver metastasis progression, which correlated with reduced NET production [112].

### 6.4. Study Design Issues

There are challenges in designing clinical trials in a way that accurately reflects both clinical improvements and NET inhibition. Firstly, unlike inflammatory markers such as CRP, there is a lack of standardised biomarkers that reliably quantify NET levels in vivo. The current biomarkers for NETs have their limitations. A commonly used biomarker is cell-free DNA (cfDNA), and while NETs are made up of DNA, they can also result from apoptotic and necrotic cells, making the marker non-specific [157,158]. MPO-DNA complexes can be more specific for NETs; however, MPO is also released from degranulation and can form complexes with cfDNA released due to apoptosis or necrosis. Additionally, the most common way to measure MPO-DNA is through an ELISA, which was found to be error-prone when measuring NET information [159]. Citrullinated histone H3 is a PAD4-dependent marker that fails to account for the PAD4-independent types of NET formation [107,160]. As a result, even if NET inhibition is achieved in relevant patients, it is not accurately reflected by current biomarkers, making it suboptimal to use as a surrogate endpoint in clinical trials.

## 7. Conclusions

The hundreds of preclinical studies regarding NET-targeted therapies, many of which can be considered successful, are evidence that this therapeutic strategy has potential for clinical use. However, several challenges remain in the process of translating from bench to bedside. Many repurposed drugs, while effective in preclinical models, remain unexplored or unsuccessful in clinical trial settings. This is, in part, due to the complexity and variability of NETs across different diseases, thus complicating the identification of universal therapeutic targets. In addition, using animal models and isolated cells is limited in that they do not fully capture the intricacies of NET-associated pathology in humans. Dosing and safety concerns also pose significant hurdles, as excessive dosage of any kind of immunosuppressive drug leaves patients highly susceptible to infections. Furthermore, the development of reliable biomarkers for patient stratification will be essential for refining these therapeutic strategies. Overcoming these challenges will require improved translational models and well-designed clinical trials to balance efficacy with safety in NET-targeted therapies.

## Figures and Tables

**Figure 1 ijms-26-04773-f001:**
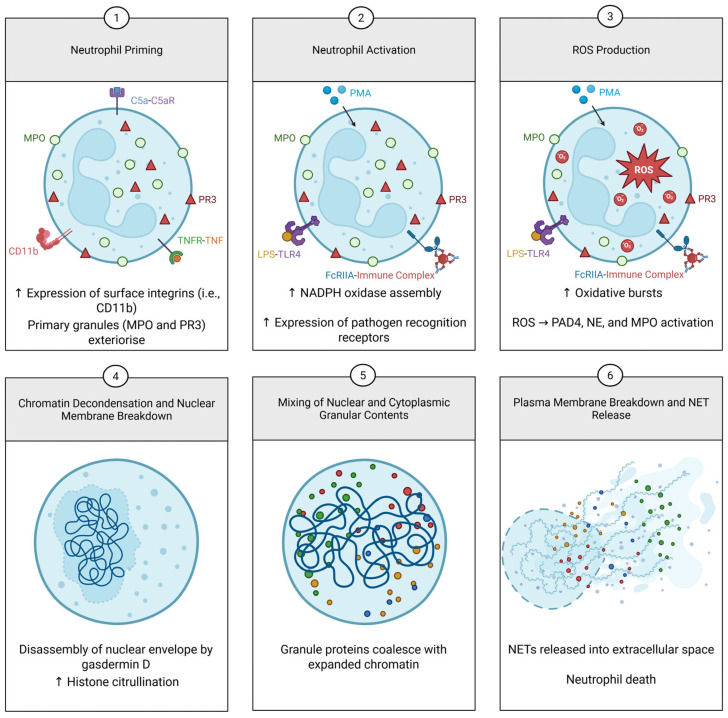
Overview of the canonical formation of neutrophil extracellular traps (NETs). Resting neutrophils are primed by various agents, such as C5a and tumour necrosis factor (TNF). This causes azurophilic granules (i.e., myeloperoxidase (MPO) and proteinase 3 (PR3)) to be exteriorized and expression of surface adhesion molecules (i.e., CD11b) to be upregulated, leading to a reduction in the threshold for activation. Stimulants, such as phorbol 12-myristate 13-acetate (PMA), lipopolysaccharide (LPS), and immune complexes, activate neutrophils, resulting in upregulation of pathogen recognition receptors (PRRs) and NADPH oxidase assembly, which then leads to increased production of reactive oxygen species (ROS). ROS drives the activation of peptidyl arginine deiminase 4 (PAD4), neutrophil elastase (NE), and MPO, which results in chromatin decondensation and nuclear breakdown. Nuclear contents amalgamate with cytoplasmic granules, and the resulting NET is released upon plasma membrane breakdown. ↑ indicates an increase in the corresponding cellular process. → indicates activation of the corresponding components. Available at: https://BioRender.com/ebuzdn7 (accessed on 7 April 2025).

**Figure 2 ijms-26-04773-f002:**
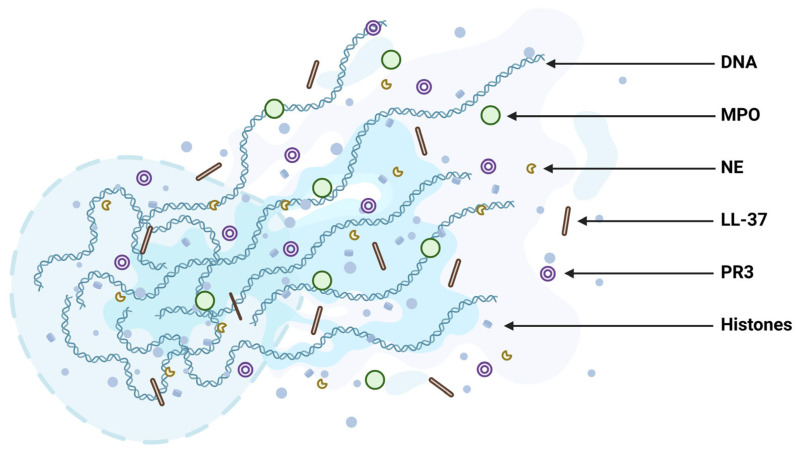
Structure of neutrophil extracellular traps (NETs). NETs are composed of a double-stranded DNA backbone that is web-like in structure. This backbone serves as a scaffold for an array of proteins, including histones (H1, H2A, H2B, H3, and H4), often modified (e.g., citrullination). They are also coalesced with antimicrobial proteins like neutrophil elastase (NE), myeloperoxidase (MPO), proteinase 3 (PR3), and LL-37. Available at: https://BioRender.com/ree6pjh (accessed on 7 April 2025).

**Figure 3 ijms-26-04773-f003:**
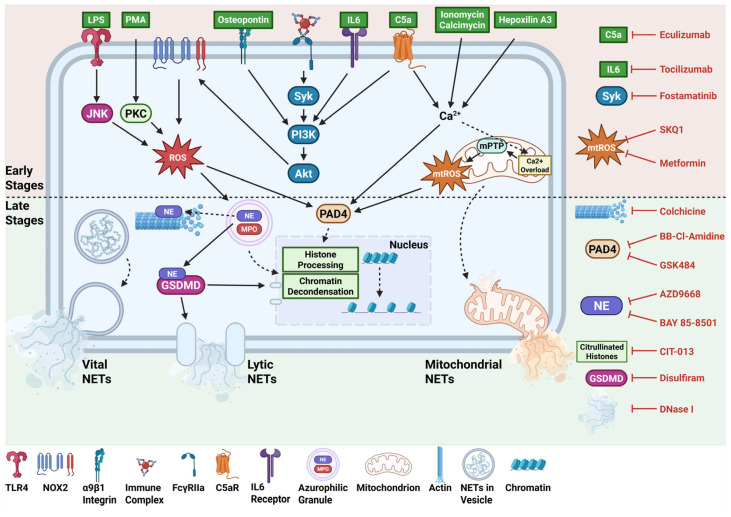
Pathways of neutrophil extracellular trap (NET) formation and targets of NET inhibitors. Neutrophils are activated by calcium ionophores ionomycin and calcimycin, C5a-C5a receptor (C5aR) interactions, and Hepoxilin A3, all of which lead to an increase in intracellular Ca^2+^. Ca^2+^ activates peptidyl arginine deiminase 4 (PAD4), which translocates to the nucleus and citrullinates histones, aiding in chromatin decondensation. Ca^2+^ can also act on mitochondria, promoting mitochondrial neutrophil extracellular trap (NET) release and the production of mitochondrial reactive oxygen species (mtROS), which can activate PAD4. Immune complexes bind to Fc gamma receptor IIa (FcγRIIa) and activate the Syk-PI3K-Akt pathway, leading to the activation of NADPH oxidase 2 (NOX2) and generation of ROS. ROS activates PAD4, as well as neutrophil elastase (NE) and myeloperoxidase (MPO) in azurophilic granules, which aid in histone processing and chromatin loosening. NE also promotes actin degradation and activation of gasdermin D (GSDMD) that forms pores in nuclear and plasma membranes for DNA extrusion. Stimulation with osteopontin, interleukin-6 (IL-6), and C5a directly activates PI3K, leading to NOX2-dependent NET release. LPS, recognised by TLR4, leads to activation of c-Jun-N-terminal kinase (JNK) and, consequently, NOX2-derived ROS. Similarly, direct activation of protein kinase C (PKC) by Phorbol 12-Myristate 13-Acetate (PMA) induces lytic NETosis via NOX-2 ROS. Several drugs target these pathways, such as eculizumab (C5a), tocilizumab (IL-6), fostamatinib (Syk), SKQ1 and metformin (mtROS), colchicine (actin), BB-Cl-Amidine and GSK484 (PAD4), AZD9668 and BAY 85-8501 (NE), CIT-013 (citrullinated histones), disulfiram (GSDMD), and DNase I, to enhance NET clearance. Available at: https://BioRender.com/hkw427v (accessed on 7 April 2025).

**Table 1 ijms-26-04773-t001:** Preclinical studies of NET-inhibiting drugs.

Drug	Mechanism/Target	Experimental Model/Disease Context	Key Findings	Reference
Eculizumab	Binds to the complement component C5 and inhibits its cleavage	In vitro, plasma from paroxysmal nocturnal haemoglonuria patients with or without a history of thrombosis	Reduced nucleosome levels in the thrombosis group, suggesting reduced NET formation.	[64]
Fostamatinib	Inhibits Syk, reducing FcR-mediated NET formation	In vitro, plasma from COVID-19 patients	Reduced NET release by healthy neutrophils stimulated with plasma from COVID-19 patients	[65]
SKQ1	Scavenges excess mitochondrial reactive oxygen species	In vitro, isolated neutrophils from healthy patients or those with CGD	Inhibited calcium-induced NETs but not PMA-induced NETs	[46]
Colchicine	Disrupts microtubule mobilisation. Inhibits ROS and calcium influx	In vivo, C57Bl/6 mice subjected to ligation of the left anterior descending coronary artery	Improved survival and cardiac function, and inhibited NET formation	[66]
BB-Cl-Amidine	Inhibits PAD enzymes, limiting histone citrullination	In vivo, lupus-prone MRL/*lpr* mice	Improved endothelial function and downregulated type I IFN-regulated genes while decreasing NET formation	[67]
GSK484	Inhibits PAD4 enzymes, limiting histone citrullination	In vitro, isolated murine neutrophils	H3Cit^+^ cells and NETs were reduced significantly	[68]
CIT-013	Binds to citrullinated histones, preventing release of NETs	In vivo, the CAIA mouse model, the DSS-induced colitis mouse model, and the LPS-induced sepsis mouse model	For all disease models, severity was reduced, and NET release diminished	[69]
Disulfiram	Inhibits gasdermin D, blocking plasma membrane rupture	In vivo, the transfusion-related acute lung injury mouse model and the COVID-19 hamster model	NET formation was blocked in both models. Survival increased in ALI mice, and lung histology improved in both models.	[70]
NINJ1 Monoclonal Antibodies	Blocks NINJ1—a plasma membrane protein required for rupture	In vivo, the acute oxalate nephropathy mouse model	Deletion of NIN1 reduced NETs, inflammation, and renal damage	[71]
DNase I	Degrades the extracellular DNA backbone of NETs, promoting clearance	In vivo, the ischemia–reperfusion injury rat model	Inflammation and epithelial damage improved due to NET degradation	[72]

**Table 2 ijms-26-04773-t002:** Clinical trials of NET-inhibiting drugs.

Drug	Mechanism/Target	Trial Design	Outcomes	Reference
Tocilizumab	Binds to IL-6 receptor to prevent IL-6 signalling	Phase II trial assessing tocilizumab (280 mg, intravenous, single dose) vs. placebo during percutaneous coronary intervention in patients with STEMI	All NET markers were reduced compared to the placebo. Myocardial salvage improved	[73]
Metformin	Activates AMPK and inhibits mitochondrial complex 1, reducing mtROS	Sub-study from a phase III trial assessing metformin (1500 mg/day for 2 months) vs. placebo in pre-diabetic patients	After 2 months of therapy, metformin significantly reduced NET markers compared to placebo	[74]
AZD9668	Inhibits neutrophil elastase, limiting gasdermin D activation and histone modifications	Phase II trial assessing oral AZD9668 (60 mg, bid, for 28 days) vs. placebo in patients with idiopathic or post-infective bronchiectasis	No significant improvements in sputum neutrophil counts or lung function. AZD9668 was well-tolerated without serious adverse events.	[75]
AZD9668	-	Phase II trial assessing oral AZD9668 (60 mg, bid, for 28 days) with standard care vs. placebo with standard care in cystic fibrosis patients	No significant improvements in sputum neutrophil counts or lung function. AZD9668 was well-tolerated without serious adverse events.	[76]
AZD9668	-	Phase I/II trial assessing oral AZD9668 (5, 20, and 60 mg, bid, for 12 weeks) vs. placebo in COPD patients receiving tiotropium treatment	No difference in post-bronchodilator FEV1 compared to placebo. No difference in adverse events relative to placebo	[77]
BAY 85-8501	Inhibits neutrophil elastase, limiting gasdermin D activation and histone modifications	Phase II trial assessing BAY 85-8501 (1 mg/day for 28 days) vs. placebo in patients with non-CF BE	There was no improvement in the disease. There was no statistical difference between the metformin and placebo groups regarding adverse effects	[78]
DNase I	Degrades the extracellular DNA backbone of NETs, promoting clearance	Phase II trial assessing nebulised DNase I (2.5 mg, days 1–7) with best available care (BAC) vs. BAC alone in hospitalised COVID-19 patients	CRP and D-dimer were reduced, indicating reduced inflammation through NET degradation	[79]

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
