# Peer review of "A Web of Challenges: The Therapeutic Struggle to Target NETs in Disease"

_ijms, 2025, doi:10.3390/ijms26104773_

Round 1
Reviewer 1 Report
Comments and Suggestions for Authors
A great review that should be published. However, it can be improved.
My suggestions are:
- add more schemes - at least one that summarizes the content/structure of a NET described in the text
- add more details about biochemical changes to the NETs and affect their stability and immunogenicity - oxidation, glycation - which might have consequences on the role of NETs in aging and the lack of effects of DNases
- I would like to know more about the preclinical and clinical studies targeting NETs - tables summarizing the available data would be very valuable - not only the drug, but also its effects and details about the studies - or that it is likely needed to divide animal and human studies.
Author Response
- Add more schemes - at least one that summarizes the content/structure of a NET described in the text. Thank you for this suggestion we have now added 2 extra figures showing how a NET is formed from activation, decondensation of DNA and release (figure 1 on page 2), and another showing the structure of NETs and the proteins embedded in them (figure 2 on page 3).
- Add more details about biochemical changes to the NETs and affect their stability and immunogenicity - oxidation, glycation - which might have consequences on the role of NETs in aging and the lack of effects of DNases
Thank you for this recommendation. We have added an extra section specifically talking about biochemical modifications to NETs and have included both oxidation and glycation. This can be found on page 8.
- I would like to know more about the preclinical and clinical studies targeting NETs - tables summarizing the available data would be very valuable - not only the drug, but also its effects and details about the studies - or that it is likely needed to divide animal and human studies.
We agree that this would greatly improve clarity and utility for readers. Hence, we have modified the table, adding more details and splitting it into preclinical (table 1) and clinical trials (table 2), which are found on page 9 and 10, respectively.
Reviewer 2 Report
Comments and Suggestions for Authors
This MS is well written and the current data conserning the role of NETs are clearly presented. Thus, this manuscript can be published in the present form.
Currently, the correlation between NETs and the pathogenesis of multiple diseases (i.e., autoimmune diseases, inflammatory diseases, cancer) have become an attractive therapeutic target. However, none of the preclinical studies inhibiting NETs with various drugs have been approved for clinical use. The topic of this review addresses this specific gap and tries to answer why, despite the enormous potential of NET inhibitors, NET-targeted therapies are failing to reach clinical approval. The authors discuss the different approaches to targeting NETs, the current clinical trials on NET-based therapies, and the biggest challenges in translational therapy surrounding NETs.
The references are appropriate. The Figure 1. Is OK.
The conclusions of the authors consistent with the evidence and arguments presented and may explain what is required to go from bench to bedside successfully.
Author Response
This MS is well written and the current data conserning the role of NETs are clearly presented. Thus, this manuscript can be published in the present form.
Currently, the correlation between NETs and the pathogenesis of multiple diseases (i.e., autoimmune diseases, inflammatory diseases, cancer) have become an attractive therapeutic target. However, none of the preclinical studies inhibiting NETs with various drugs have been approved for clinical use. The topic of this review addresses this specific gap and tries to answer why, despite the enormous potential of NET inhibitors, NET-targeted therapies are failing to reach clinical approval. The authors discuss the different approaches to targeting NETs, the current clinical trials on NET-based therapies, and the biggest challenges in translational therapy surrounding NETs.
The references are appropriate. The Figure 1. Is OK.
The conclusions of the authors consistent with the evidence and arguments presented and may explain what is required to go from bench to bedside successfully.
Thank you for your comments, we have now added two extra figures and a table outlining the different roles of inhibitors from both mouse and clinical trials.